# Generating Errors: OCR Post-Processing for Icelandic

**Atli Jasonarson, Steinþór Steingrímsson, Einar Freyr Sigurðsson,**
**Árni Davíð Magnússon, Finnur Ágúst Ingimundarson**
The Árni Magnússon Institute for Icelandic Studies, Iceland
`{atli.jasonarson, steinthor.steingrimsson,`
`einar.freyr.sigurdsson, arni.d.magnusson}@arnastofnun`
`fai@hi.is`

## Abstract

We describe work on enhancing the performance of transformer-based encoder-decoder models for OCR post-correction on modern and historical Icelandic texts, where OCRed data are scarce. We trained six models, four from scratch and two fine-tuned versions of Google's ByT5, on a combination of real data and texts populated with artificially generated errors. Our results show that the models trained from scratch, as opposed to the fine-tuned versions, benefited the most from the addition of artificially generated errors.

## 1 Introduction

Optical Character Recognition (OCR) is used to digitize texts by converting scanned documents into machine-readable text. Unfortunately, OCR errors are prevalent, particularly when it comes to old texts, where data tends to be scarce, and post-correction is often required to improve the extracted texts' accuracy (e.g. Nguyen et al. 2021).

Transformer-based encoder-decoder models have been shown to be effective in various natural language processing tasks, including machine translation (Vaswani et al., 2017; Chen et al., 2018) and text summarization (Garg et al., 2021). In this study, we investigate the use of such models for OCR post-correction under scarce data condition, as a sequence-to-sequence problem, similar to how neural machine translation (NMT) systems approach the problem of translation. To address the lack of resources available for training models when dealing with OCRed texts, we propose the use of artificially generated errors to improve the performance of the models, which has been shown to be an effective way of generating data for text correction (Kasewa et al., 2018). The main contribution of this study is an examination of the effectiveness of using artificially

generated errors to improve the performance of transformer-based encoder-decoder models for OCR post-correction when data scarcity is a limiting factor. Furthermore, we publish our best performing models under the Apache 2.0 license.[1]

The paper is structured as follows. Section 2 discusses related work while Section 3 describes the dataset and error generation methods used in this study. Section 4 presents the proposed models and training methods. Section 5 presents the experimental results and analysis, Section 6 discusses limitations and future work, and finally Section 7 concludes.

## 2 Related Work

Previously, Daðason et al. (2014) developed a tool for post-processing Icelandic 19[th] century texts based on an error model containing statistical information on word and character errors and an n-gram language model. Their tool correctly identifies and corrects 52.9% of errors in their evaluation set.

Poncelas et al. (2020) report a 63% error correction rate with their OCR post-processing tool on an English text from the 18[th] century. They used a scoring system based on string-similarity to find possible substitutions for perceived errors and a language model to evaluate the edited sentences.

Richter et al. (2018) use a hidden Markov model alongside a modified version of the Viterbi algorithm and a dictionary to decode OCRed texts in Faroese into a hypothetical corrected version. They reduced the word error rate of 7.8% from the OCR base process to 5.4%.

---

[1]Models available at the following URLs: http://hdl.handle.net/20.500.12537/271, http://hdl.handle.net/20.500.12537/309, https://huggingface.co/atlijas/byt5-is-ocr-post-processing-modern-texts, https://huggingface.co/atlijas/byt5-is-ocr-post-processing-old-texts.

| Original | Corrected | Frequency |
|----------|-----------|-----------|
| *p* | *þ* | 2,779 |
| *i* | *í* | 1,141 |
| *li* | *h* | 247 |
| *rn* | *m* | 166 |
| *m* | *rn* | 77 |

Table 1: Examples of the extracted errors.

| | Model 1 | Model 2 |
|---|---------|---------|
| embeddings size | 512 | 512 |
| ffn embeddings | 2,048 | 2,048 |
| attention heads | 4 | 4 |
| encoder layers | 5 | 5 |
| decoder layer | 5 | 5 |
| tokenizer | WordPiece | SentencePiece |
| vocab. size | 3,000 | 3,000 |

Table 2: The architecture of the two models trained from scratch.

## 3 Data

We used a combination of real OCRed texts, processed by ABBYY FineReader,[2] and digital texts not scanned with OCR, the latter of which were populated with artificially generated errors, for training and evaluating our OCR post-correction models. The evaluation data solely comprised real OCRed texts, alongside their manually corrected counterparts, which were used to ensure that the models' performance is reflective of real-world OCR output. The ground truth (GT),[3] i.e. the data from which the errors were extracted, consists of around 375k tokens from 80 texts published between 1874 and 1913, which were manually corrected, while the data used in training and validation, which include the OCRed texts as well as the texts populated with the artificial errors, amount to roughly 9.2M tokens.

The data into which the artificial errors were inserted were taken from the Icelandic Gigaword Corpus (IGC; Steingrímsson et al. 2018) and the Icelandic Text Archive (ITA).[4] Their publication dates range from the late 18$^{\text{th}}$ century to the early 21$^{\text{st}}$ century, with roughly 40% of them having been published between 1830 and 1920.

Overall, the training data consist of 7.8M tokens, whereas the validation set, which is approximately 15% of the whole dataset, consists of around 1.4M tokens. The evaluation set, totalling 44k tokens, is composed of manually corrected texts. All datasets contain texts from different eras, including texts from the 19$^{\text{th}}$ century and the early 20$^{\text{th}}$ century, as well as texts from the last two decades of the 20$^{\text{th}}$ century. It should be noted that none of the data are based on texts printed in Gothic font, which has been reported to be harder to recognize than other fonts (Furrer and Volk, 2011; Drobac et al., 2017). The evaluation set is divided into two parts, with 26k tokens being from modern texts and 18k tokens from texts from the 19$^{\text{th}}$ century and the early 20$^{\text{th}}$ century. This allows for an evaluation of the model's performance on different types of texts and OCR errors, which is crucial to ensure that the model is robust and generalizable.

It is important to note that the dataset used in this study is relatively small in size. One of the reasons is the scarcity of available corrected OCRed texts. Additionally, we observed that too large a proportion of modern texts in the training set resulted in the models over-generalizing and changing historical spellings to modern spellings. However, we aim for diplomatic transcription, preserving the original spelling. Therefore, we ensured that the dataset included texts from different eras while also avoiding over-generalization and alteration of historical spellings by limiting the amount of modern texts into which we inserted artificial errors.

### 3.1 Extracting the Errors

The extraction of errors from the manually corrected OCRed texts was performed by analysing the 375k token dataset. The data were manually aligned, and then a line-by-line comparison was conducted between the OCRed texts and their corresponding manually corrected texts using Python's SequenceMatcher. In the process of extracting errors, tokens were considered to be the same if they shared the same index in a given line and had a similarity score greater than 0.66.[5] This twofold restriction, taking into account both index

[2]https://pdf.abbyy.com/

[3]The GT is a product of the project *Language Change and Linguistic Variation in 19$^{th}$-Century Icelandic and the Emergence of a National Standard*, led by Ásta Svavarsdóttir at the Árni Magnússon Institute for Icelandic Studies (e.g. Svavarsdóttir et al. 2014).

[4]https://clarin.is/en/resources/textarchive/

[5]Calculated by finding "[...] the longest continuous matching subsequence that contains no "junk" elements", see: https://docs.python.org/3/library/difflib.html.

|  | Older texts | | | | Modern texts | | | |
|---|---|---|---|---|---|---|---|---|
|  | OCR | Model 1 | Model 2 | ByT5 (5 ep.) | OCR | Model 1 | Model 2 | ByT5 (1 ep.) |
| chrF | 94.79 | 94.80 | 96.00 | 96.22 | 95.27 | 95.52 | 95.75 | 96.09 |
| BLEU | 97.19 | 97.19 | 98.24 | 98.54 | 97.73 | 97.63 | 98.06 | 98.24 |
| WER | 6.49% | 7.56% | 4.22% | 3.25% | 5.52% | 5.73% | 4.58% | 4.56% |
| WERR | Ø | -16.37% | 35.04% | 49.96% | Ø | -3.80% | 17.02% | 17.37% |
| CER | 1.39% | 1.79% | 1.14% | 0.92% | 1.17% | 1.63% | 1.43% | 1.41% |
| CERR | Ø | -28.53% | 18.34% | 33.83% | Ø | -38.58% | -21.34% | -20.34% |

Table 3: Our models trained on the GT compared to the base output from the OCR process. WER(R) = Word Error Rate (Reduction), CER(R) = Character Error Rate (Reduction).

and similarity, acted as a confidence threshold to ensure that the identified tokens were different versions of the same intended token.

The differences between the tokens, specifically focusing on character or character n-gram substitutions, such as *rn→m* and *þ→p*, were extracted as OCR errors. In total, 2,644 such error types were extracted, which were then filtered down to the 600 errors that occurred more than three times in the dataset. In addition, the frequency of each error was recorded, which allowed for the implementation of a weighting system during the artificial error generation process, ensuring that the errors were distributed in a way that somewhat reflected their real-world frequency. Examples of extracted errors are shown in Table 1.

Error pairs that consist of an original and corrected string length 1 (character count) comprise around 40% of the error set. An example of this is the erroneous *pessi* for *þessi* 'this'. In 30% of error pairs the original has a length of 2 and the corrected a length of 1, such as *rnaður* for *maður* 'man', and in about 15% of them the length of both is 2, e.g. *gdbur* for *góður* 'good'. The total number of errors in the dataset amounts to 27,369.

### 3.2 Inserting the errors

To create the training dataset, we gathered texts from IGC and the ITA. Texts ranging from the late 18$^{\text{th}}$ century to the 21$^{\text{st}}$ century were collected to provide a diverse set of texts for model training.

Error types, as extracted and described in Section 3.1, were then inserted into the training dataset by randomly replacing characters or character n-grams via a lookup table. Whitespace was also removed from between tokens and added into the tokens at random. The artificial errors were inserted randomly, with the frequency of error types

based on the frequency in the GT in order to mimic the distribution of errors that occur in OCR output. This way, more frequent errors in the GT were made to appear more frequently than other errors in the training dataset. However, to prevent the same errors from appearing excessively often in the dataset, we used the $\log_{10}$ frequency of the errors.

## 4 Models

In total, six models were trained. Two of them follow the architecture of model 1, laid out in table 2, two of them follow the architecture of model 2 in the same table, and the others are a fine-tuned version of ByT5-base[6] (Xue et al., 2022), a token-free transformer model that operates directly on UTF-8 encoded bytes and is trained on mC4, a multilingual corpus, which consists of texts in 101 languages, including Icelandic (Xue et al., 2021). The models are all encoder-decoder transformer models.

For every pair of the models, one was trained on the 375k tokens in the GT, and the other one on the whole dataset, around 7.8M tokens. This was done to study the artificially generated errors' impact on the models' output. We experimented with various hyperparameter configurations, evaluating the models we trained from scratch on the validation sets, and these specific configurations resulted in the highest performance.

It is well established that transformer models require large amounts of data to be trained effectively. In this study, our GT had a limited number of examples, which likely contributed to the

---

[6]Note that the ByT5 model was trained for five epochs, resulting in five different models. The one trained for one epoch performed the best on modern texts while the one trained for five epochs performed the best on older texts. We report on these two ByT5 models in Table 3.

poorer performance of the models trained from scratch on the smaller dataset, in some instances even performing worse than the base OCR process.

## 4.1 Tokenizers

As seen in Table 2, the two models trained from scratch use different tokenizers, both of which are based on subword tokenization algorithms. Model 1 uses WordPiece (Song et al., 2021) and model 2 uses SentencePiece (Kudo and Richardson, 2018). As mentioned before, the ByT5 model operates directly on UTF-8 encoded bytes.

Different tokenization algorithms can have an impact on a given task. SentencePiece and WordPiece can produce different subword units for the same text, which might affect the models' ability to capture language-specific nuances and patterns. It is possible that the choice of tokenizer had some impact on their performance. However, further research would be needed to determine the specific effects of the tokenizer choice on OCR error correction.

## 5 Results

The six models we trained for post-processing of OCRed texts were applied to modern and historical texts to measure the impact and viability of using artificial errors to improve such models when available data are scarce. The results of these models were compared to the base output from the OCR process using four metrics: chrF (Popović, 2015) and BLEU (Papineni et al., 2002), character error rate (CER) and word error rate (WER). BLEU score is calculated by comparing texts on a word-level, while chrF score is calculated on a character-level and can be more accurate for inflected languages (Dowling et al., 2020).

Table 3 shows the results of our models trained on the GT compared to the base OCR output. Model 2 and the ByT5 model show moderate improvements for older texts, while model 1 performs similarly or worse than the base OCR output. Generally, the models do worse on the modern texts, as opposed to the historical ones, when only trained on the GT, which is to be expected as the GT solely consists of historical texts.

Table 4 shows the results of our models trained on the whole dataset compared to the base OCR output. The models all show substantial improvements compared to the models only trained on the

GT, which suggests that the artificial errors have something to offer. Furthermore, the difference between word error rate reduction (WERR) for the different text types was less than for the models only trained on the GT.

Note that while the models generally perform better on the historical texts, the addition of artificial errors improve their performance proportionally more on the modern ones. This could stem from the fact that the artificially-erroneous dataset includes modern texts, while the GT does not.

When evaluating the models on modern texts, we found that they were less capable in reducing errors in modern texts than in historical texts. This could be due to the fact that the GT only comprised historical data, suggesting that using solely historical OCRed texts is not a viable approach when designing an OCR post-processing tool for modern texts. The lower error rate reduction (ERR) on the modern texts presumably also stems from the higher base score on the modern texts, as opposed to the base score of the historical ones, leaving less room for improvement.

## 6 Limitations and Future Work

The cost of manually correcting OCR output is high, making it difficult to obtain a larger dataset for training. This has a direct impact on the ability of the models to perform well on a wider range of texts and OCR errors.

The models have the unwanted tendency to adapt to modern spellings when using a large amount of modern texts populated with artificial errors. This could lead to the alteration of historical spellings, which is not in line with our objectives, to produce diplomatic transcriptions. To mitigate this risk, more corrected texts are needed for the period of texts being OCRed.

Moreover, the use of artificially generated errors to enhance the performance of the models may not fully capture the complexity and diversity of real-world OCR errors. Future studies may benefit from incorporating a more diverse range of error types and more realistic error generation methods.

We are interested in investigating optimal methods for generating realistic errors to use in training the models. As previously mentioned, the artificial errors used in this study were generated by randomly inserting errors that were extracted from the GT into other texts. However, there may be more effective methods for generating errors that better

|  | Older texts | | | | Modern texts | | | |
|---|---|---|---|---|---|---|---|---|
|  | OCR | Model 1 | Model 2 | ByT5 (5 ep.) | OCR | Model 1 | Model 2 | ByT5 (1 ep.) |
| chrF | 94.79 | 96.84 | 96.84 | 96.73 | 95.27 | 96.83 | 96.86 | 96.7 |
| BLEU | 97.19 | 98.45 | 98.79 | 98.65 | 97.73 | 98.45 | 98.64 | 98.57 |
| WER | 6.49% | 4.95% | 3.08% | 2.92% | 5.52% | 4.52% | 3.60% | 3.15% |
| WERR | Ø | 23.79% | 52.60% | 55.07% | Ø | 18.00% | 34.67% | 42.97% |
| CER | 1.39% | 1.03% | 0.73% | 0.90% | 1.17% | 1.06% | 1.0% | 1.15% |
| CERR | Ø | 26.29% | 47.55% | 35.12% | Ø | 10.01% | 15.20% | 1.93% |

Table 4: Our models trained on the whole dataset compared to the base output from the OCR process.

simulate real-world OCR errors. By finding and implementing these methods, the performance of OCR error correction models could be further improved. Furthermore, it could be beneficial to explore different architectures or different data augmentation techniques, such as including multiple versions of the same texts. It should also be noted that our evaluation dataset was rather small, and further testing on larger datasets may provide a more robust evaluation of the models.

## 7 Conclusion

Our findings demonstrate that while fine-tuning pre-trained models on smaller datasets is an effective approach to improving the performance of OCR error correction models, it is possible to achieve comparable results by training an encoder-decoder transformer model from scratch. Model 2, which was trained from scratch, emerged as the best performer in our study, achieving a 52.60% word error rate reduction (WERR) and a 47.55% character error rate reduction (CERR) on the historical texts, and a word error rate reduction of 34.67% and a character error rate reduction of 15.20% on the modern texts, see Table 4.

These results indicate that with proper architectural design, it is possible to train effective OCR error correction models without relying on pre-trained models or large datasets.

However, the use of artificially generated errors in the training process was found to be effective in countering the challenges posed by data scarcity.

The fact that the models' performance improved proportionally more on the modern texts after the introduction of the artificial errors, which were by and large inserted into modern texts, indicates that in order to train a designated OCR post-processing tool for modern texts, a dataset consisting of modern texts is needed.

## Acknowledgements

This work is supported by the Language Technology Programme for Icelandic 2019–2023, funded by the Icelandic government, and the Icelandic Infrastructure Fund (grant no. 200336-6101).

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
