# OpenReview forum: "Generating Errors: OCR Post-Processing for Icelandic"
_NoDaLiDa/2023/Conference — NoDaLiDa 2023_

### Official Review · Reviewer_g7FH · 2023-02-25
**The authors generated errors to extend their training set for obtaing better OCR recognition for Icelandic historical texts but also for modern Icelandic texts. They obtained better results with their approach than other previous approaches.**

**Rating:** 7
**Confidence:** 3

**Review:**


What is ByT5, how does it work, what type of models is it?
Is it pretrained on Icelandic?

Model 1 and Model 2 what type of models are they? BERT based or what?

From what years are the historical icelanding texts?
Are they contained in IGC and ITA corpora?

You obtained better results than (Daðason et al. 2014)
but also better results than (Poncelas et al. 2020) that worked on 18th century English text but you obtained worse results than
(Richter et al. 2018) for Faroese. Have you thought about using the method of (Richter et al. 2018)?


Some smaller comments:

ABBYY FineReader => please add reference.

Some English comments

Section 4 presents the proposed
models and training methods. Section 5 presents
the experimental results and analysis, 6 discusses
limitations and future work and 7 concludes.
=>
Section 4 presents the proposed
models and training methods. Section 5 presents
the experimental results and analysis, Section6 discusses
limitations and future work and finally Section 7 concludes.





**Paper Type:**

Short paper

---

### Official Review · Reviewer_rQpC · 2023-03-06
**A spellchecker specialised to OCR errors, for Icelandic.**

**Rating:** 7
**Confidence:** 4

**Review:**



This work is not particularly ground breaking, and unfortunately it
does not seem to lead to any particularly new insight, even after the
improvements that I can imagine.  However the paper reports on a fair,
self contained nugget of research which fits appropriately in the
scope of the conference.


clarity:
 - please explain what an "Error" is. It is a pair (input,output)
   which is incorrect? If so, how training on incorrect data improve
   the model's performance?  It it the case that the model should not
   only recognize the glyphs but also correct them according to
   context when they are incorrectly drawn?  You should clarify this
   immediately in the intro.

 - As far as I can ascertain, you're considering a post-processing
   step on top of a fixed OCR model. You're not actually improving an
   OCR model. This needs to be made plain in the introduction.

   But then, you're effectively adding a spellchecker to the
   pipeline. (In this light, it's hardly surprising that your model
   "corrects" archaic spellings) What you appear to be doing is
   training a spellchecker which is attuned to the kind of errors
   found in OCR.

 - I don't think that you ever define precisely the "base OCR" model
   that you refer to in the tables.

quality:
 - This is fair work, though I'd like the authors to clarify how (and
   how much) they tuned the hyperparameters.

originality:
 - If I'm correct this is a fine tuning of a spellchecker, which isn't
   particularly original.

significance:
 - relatively niche (but so is the scope of the venue)
 - The GT does not appear to be part of this work (no bonus here)
 - It's good that the models are made available with open license.


Details:


103: ABBYY FineReader: missing reference

144: fonts -> alphabet  (are you really talking about a font for the Latin
alphabet or is it another alphabet? Seems you mean the latter since
you refer to "Gothic letters" earlier in the sentence)

156: "too a large proportion" -> "training on a too large proportion" (?)

267: "frequency" fix typesetting (\mathit)

256: "raw text" -> UTF8 encoding of text (?)

257: "every pair of the models" -> "every pair of models"

265: it is good that you're honest about twiddling with
hyperparameters. Unfortunately without more information (how thorough
have you been, even qualitatively) it's hard not to put your results
in doubt.


Tables 3. I find it quite hard to make sense of this table at first
reading. There appears to be both absolute accuracies and relative
percentage numbers. For "chrF ERR", what you show is a percentage
difference on what is already is a proportion of errors (but it could
be percentage points).  I suggest you only show the *absolute* scores
or error rates for each metric. If you absolutely want to insist on
the relative effects, show them separately and indicate cleary what
they mean. (Likewise Table 4)

430: do you mean to replace "modern" by "ancient" in the last two
occurences of the sentence?



**Paper Type:**

Short paper

---

### Official Review · Reviewer_SJsm · 2023-03-10
**Improving the OCR for Icelandic by inserting data with inserted artifical errors**

**Rating:** 8
**Confidence:** 5

**Review:**

The paper presents work on applying transformer-based encoder-decoder models for post corrections of OCR errors for Icelandic text.

Strengths: State-of-the-art transformer based models are tested on the complex task of improving the accuracy of historical and modern texts containing OCR errors. The dataset for training and testing was increased by inserting artificial errors. The models and insights from the work are valuable contributions to the NLP community.

Weaknesses: There are no detailed specifications of the artificial errors, i.e. how many types of errors were extracted and what is the frequency of each in the data. The amount of the errors extracted from each period is not mentioned either. It is not clear whether the authors reached state-of-the-art results for Icelandic historical texts.

Detailed comments:

l. 103: Does regular texts mean not scanned text?

l. 133-139: The numbers mentioned in this paragraph might read better in a table.

l. 169: What does 'SentencePiece' mean?

l. 208: How was the weighting system implemented more specifically.

Table 3: The paper could benefit from being more self contained, for example explain how chrF is calculated.

**Paper Type:**

Short paper

---

### Decision · Program_Chairs · 2023-03-17

Accept